# Synergistic Space-Vision Processing for Predicate Inference

**Zhenhua Lei** [1]  **Zefang Han** [2]  **Yu Qiu** [2]

## Abstract

Scene graph generation (SGG) aims to parse an image into a structured graph of objects and their predicates, enabling explicit relational reasoning for visual understanding. However, prevailing methods often over-predict geometric predicates, resulting in scene graphs that are factually correct yet semantically shallow. While recent works effectively attribute this phenomenon to the long-tailed data distribution, we identify another critical factor driving such biased prediction: co-occurrence-induced representation entanglement, where geometric and non-geometric predicates that frequently co-occur are encoded into overly similar representations. To this end, we introduce Dual-stream Synergistic Network (DS-Net) that models geometric and non-geometric predicates with two specialized streams, coupled with a bidirectional cross-stream fusion mechanism. The space stream focuses on spatial and structural cues, while the vision stream captures fine-grained visual evidence and semantic priors. Extensive experiments show that DS-Net consistently improves predicate inference, achieving $1.3\% \sim 6.1\%$ absolute gains in mR@100 on the SGGen task when integrated into existing SGG methods. These results highlight the importance of synergistic modeling of geometric and non-geometric predicates for generating semantically richer scene graphs.

## 1. Introduction

Scene graph generation (SGG) aims to parse an image into a structured graph of objects and pairwise predicates, enabling explicit relational reasoning for high-level visual understanding (Shi et al., 2019; Qian et al., 2024; Lin et al.,

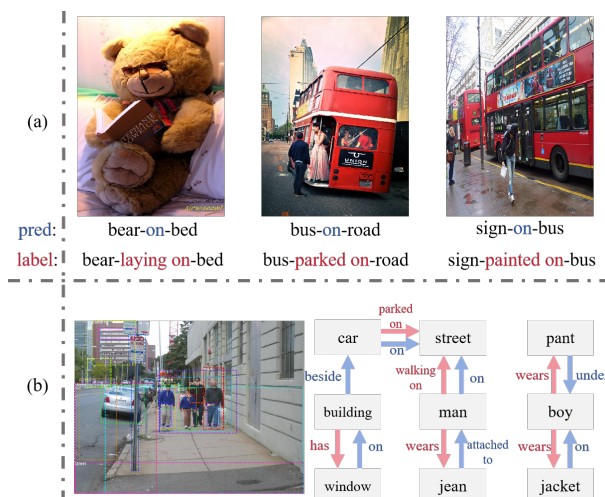

*Figure 1.* (a) Existing methods tend to over-predict the geometric predicates (e.g., on), while the ground truth corresponds to non-geometric predicates (e.g., laying on, parked on, painted on); (b) Examples of the frequent co-occurrence of geometric predicates (**blue** arrows) and non-geometric predicates (**red** arrows).

2024). Tremendous progress has been made recently in SGG, thanks to learned visual representations and advances in object detection (Chen et al., 2019; Li et al., 2024a). However, these methods often exhibit a persistent failure mode: they tend to over-predict geometric predicates (e.g., on, under, near) at the expense of semantically richer non-geometric predicates (e.g., riding, holding, eating) (Zhang et al., 2019a; Tang et al., 2019; Li et al., 2021), as illustrated in Figure 1.**a**. Consequently, the generated scene graphs are often factually correct yet semantically shallow.

Recent works commonly attribute this phenomenon to the biased prediction stemming from long-tailed data distributions, noting that geometric predicates dominate standard SGG datasets (Zhang et al., 2019a; Tang et al., 2020; Abdelkarim et al., 2021; Li et al., 2021). For example, in Visual Genome dataset (Krishna et al., 2017), geometric predicates account for nearly 50% of the data, while semantic predicates that describe the actions between objects account for only 9% (Zellers et al., 2018). This observation has led most recent works to focus on mitigating the long-tailed data distribution (Tang et al., 2020; Zheng et al., 2023b; Li et al., 2024d; Liu et al., 2025), while overlooking another

[1]School of Computer Science and Technology, Xi'an Jiaotong University, Xi'an, China [2]School of Artificial Intelligence, Xi'an University of Electronic Science and Technology, Xi'an, China. Correspondence to: Zhenhua Lei <leizh.yh@foxmail.com>.

*Proceedings of the 43rd International Conference on Machine Learning*, Seoul, South Korea. PMLR 306, 2026. Copyright 2026 by the author(s).

intriguing fact: the inherently heterogeneous geometric and non-geometric predicates frequently co-occur on the same object pair (as illustrated in Figure 1.**b**).

This phenomenon implies that the visual appearance of an object pair simultaneously encapsulates cues for both geometric and non-geometric predicates. However, for an object pair, most existing methods extract only a unified representation to infer predicates (Xu et al., 2017; Zheng et al., 2023a;b; Yoon et al., 2024). This design results in co-occurring geometric and non-geometric predicates being encoded into overly similar representations, diminishing their inter-class distinction. Coupled with the long-tailed data distribution, cues facilitating geometric inference dominate the predicate representation, thereby leading to the over-prediction of geometric predicates. We refer to this factor as "co-occurrence-induced representation entanglement". Fundamentally, this factor stems from the intrinsic properties of predicates, rendering our idea applicable to most existing methods.

Inspired by the above analysis, we introduce **D**ual-stream **S**ynergistic **Net**work (**DS-Net**), which reformulates the predicate modeling mechanism by centering on the synergy between geometric and non-geometric predicates. It comprises two key components. **First**, to prevent the geometric from dominating the representations, we substitute the conventional unified mechanism with two parallel streams: a **space stream** that focuses on the spatial layout and edge features of objects to predict geometric predicates, and a **vision stream** that infers non-geometric predicates through some fine-grained visual evidence and linguistic priors. This structural decoupling allows each super-type to be processed by a dedicated reasoning mechanism, thereby enhancing the discriminability of their learned representations.

**Second**, we introduce a **C**ross-**S**tream **F**usion (**CSF**) strategy that allows the two streams to exchange information and act as mutual priors. This strategy is motivated by the observation that geometric and non-geometric predicates can act as implicit priors for one another. Geometric predicates provide a reliable spatial prior that narrows the hypothesis space of non-geometric predicates (e.g., the triplet "human-on-horse" makes the predicate "ride" much more plausible than "hold"), while non-geometric predicates also constrain the likely spatial configuration (e.g., "human-eating-food" suggests that the two objects are more likely to be "near" than "under"). Technically, our CSF employs a bidirectional cross-attention strategy to facilitate the aforementioned cross-stream information interaction. Equipped with these components, DS-Net effectively mitigates co-occurrence-induced representation entanglement via a parallel structure, thereby promoting the balanced prediction of geometric and non-geometric predicates.

To verify the effectiveness of our method, following the common settings in SGG, we utilize Visual Genome (Krishna et al., 2017), Open Images V6 (Kuznetsova et al., 2020) and GQA (Hudson & Manning, 2019) for our experiments. Extensive results and a series of ablation experiments demonstrate the effectiveness of our ideas. Furthermore, our approach exhibits strong versatility and can be efficiently integrated with various existing models to enhance their performance. For instance, the **1.3%** $\sim$ **6.1%** increase in mR@100 on SGGen task demonstrates this capability. Our main contributions are summarized as follows:

- We identify co-occurrence-induced representation entanglement as an important factor contributing to geometric predicate over-prediction in SGG, complementary to long-tailed data imbalance.

- We propose DS-Net, a dual-stream architecture that combines specialized processing with synergistic cross-stream fusion to improve predicate representation discriminability, thereby promoting the balanced prediction of geometric and non-geometric predicates.

- Our method achieves competitive or state-of-the-art performance on various scene graph benchmarks. More importantly, our idea is model-agnostic and can be applied to several existing SGG models.

## 2. Related Works

Most existing SGG methods follow a predicate modeling paradigm, in which predicates are predicted conditioned on detected subjects and objects (Krishna et al., 2017; Tang et al., 2019; Chen et al., 2019; Zhang et al., 2019b). On top of this paradigm, many models further exploit global context and dataset statistics: motif-based architectures, tree-structured encoders, and graph-based relational networks refine features via message passing, while frequency priors derived from object–predicate co-occurrence remain strong baselines (Zellers et al., 2018; Dai et al., 2017; Li et al., 2017; Tang et al., 2019; Woo et al., 2018; Wang et al., 2019a). In this setting, predicate prediction is typically formulated as a single-head classification task with a unified feature pipeline and classifier shared across all predicates, so head categories and simple geometric predicates tend to dominate rare, semantically rich ones. Consequently, the generated scene graphs are often factually correct yet semantically shallow.

Recent works commonly attribute this phenomenon to biased prediction under long-tailed predicate distributions, noting that geometric predicates dominate standard SGG datasets (Zellers et al., 2018; Zhang et al., 2019a; Tang et al., 2020; Abdelkarim et al., 2021). These approaches aim to alleviate the imbalance by re-weighting or carefully designing loss functions (Li et al., 2021; Zheng et al., 2023b).

However, they mostly treat co-occurrence bias as a data-level problem and seldom analyze how geometric and non-geometric predicates become entangled in the representation space. In contrast, our approach explicitly targets this representation-level entanglement by decoupling geometric and non-geometric predicates into dual streams and designing a synergistic architecture that highlights and exploits their complementary cues.

## 3. Our Method

In this section, we will introduce the details of our method, and an overview of our method is illustrated in Fig.2. We first describe the foundational proposals generation process, followed by detailed descriptions of each core component.

### 3.1. Foundational Proposals Generation

**Proposal Generation.** Following standard two-stage SGG paradigms (Zellers et al., 2018; Li et al., 2021), our model operates on a set of object and relationship proposals generated from a pre-trained object detector (e.g., Faster R-CNN (Ren et al., 2015)). The object proposals are taken directly from the detection output with their categories and classification scores, while the relationship proposals are generated by forming ordered pairs of all the object proposals. Then, we compute the representations for objects.

**Object Representation Calculation.** For object representation, the calculating method follows Li et al. (2021). Specifically, for the $i$-th object proposal, we denote its convolution feature as $v_i$, its bounding box as $b_i$ and its detected class as $c_i$. Then, we can utilize $b_i$ to calculate geometric feature $g_i$, and utilize $c_i$ to calculate semantic feature $w_i$. Finally, the object representation $o_i$ is computed as

$$o_i = f_o(v_i \oplus g_i \oplus w_i), \qquad (1)$$

where $f_o$ is a MLP that integrates its visual, geometric and semantic features, and $\oplus$ is the concatenation operation.

**Two super-types of predicates.** Following the setting in Zellers et al. (2018), we partition the complete set of predicate classes into two disjoint subsets: the geometric predicates and the non-geometric predicates. The detailed division can be found in our appendix. Then, we design two parallel streams to extract the representations for them respectively, including space stream for geometric and vision stream for non-geometric.

### 3.2. Space Stream for Geometric

This stream is designed to infer geometric predicates (e.g., on, under), which are primarily determined by the spatial configuration and relative layout of objects. For this type, the edge information of the object is the most useful cue to judge it, which is represented as its bounding box obtained by object detector in the existing works (Dai et al., 2017; Li et al., 2021). While bounding box is often insufficient as it provides a coarse 2D abstraction of the 3D world, failing to capture the nuanced spatial arrangements.

To address this problem, inspired by 3D object detection, we enhance these simple edge information with visual context (Hu et al., 2018; Zhang et al., 2021; Nie et al., 2020). Specifically, we utilize an "attention sum" strategy, which refines the edge features by aggregating information from the bounding boxes and visual appearance of all other objects in the scene, effectively allowing it to learn a context-aware spatial layout (Hu et al., 2018).

For the $i$-th object proposal, this process yields an enhanced edge feature $r_{e_i}$. And then, for the relationship proposal from object $i$ to $j$, the final geometric representation $r_{g_{ij}}$ is produced by combining the enhanced edge features with the original relative spatial features:

$$r_{g_{ij}} = f_{rf}(r_{e_i} \oplus r_{e_j}) + f_g(g_i \oplus g_j), \qquad (2)$$

where $f_{rf}$ and $f_g$ are two MLPs. It allows our space stream to form a robust judgment on geometric predicates, providing a solid spatial context for the final fusion.

### 3.3. Vision Stream for Non-Geometric

This stream is designed to infer non-geometric predicates (e.g., riding, eating), which depend on some subtle visual cues and semantic priors rather than coarse spatial layouts. For example, distinguishing "hand-holding-cup" from "hand-near-cup" requires focusing on fine-grained visual cues within the interaction region. However, it is difficult and impractical to provide these fine-grained annotations for each image. To address this problem, our vision stream dynamically identifies and represents some latent parts within the union-box of an object pair, thus enhancing the attention of the model to details. It includes the following steps:

**Step.1: contextual feature compression.** Given the relationship proposal from object $i$ to $j$, we denote the convolution feature map obtained by their union-box as $u_{ij} \in \mathbb{R}^{d \times w \times h}$. We first utilize global average pooling to compress this feature map into a holistic d-dimensional vector $u_{v_{ij}}$

$$u_{v_{ij}} = gap(u_{ij}), \qquad (3)$$

where $gap$ is a global average pooling function.

**Step 2: latent part filter generation.** And then, inspired by Yang et al. (2023), we decouple the compressed feature into some latent parts. Specifically, we introduce $P$ convolutional filters, which are $P$ different MLPs: $\{\phi_p: \mathbb{R}^d \to \mathbb{R}^{d \times 1 \times 1}\}$. Every MLP independently maps the compressed feature to a $1 \times 1$ convolutional filter. Take the $p$-th MLP $\phi_p$ as the example:

$$L_{ij}^p = \phi_p(u_{v_{ij}}). \qquad (4)$$

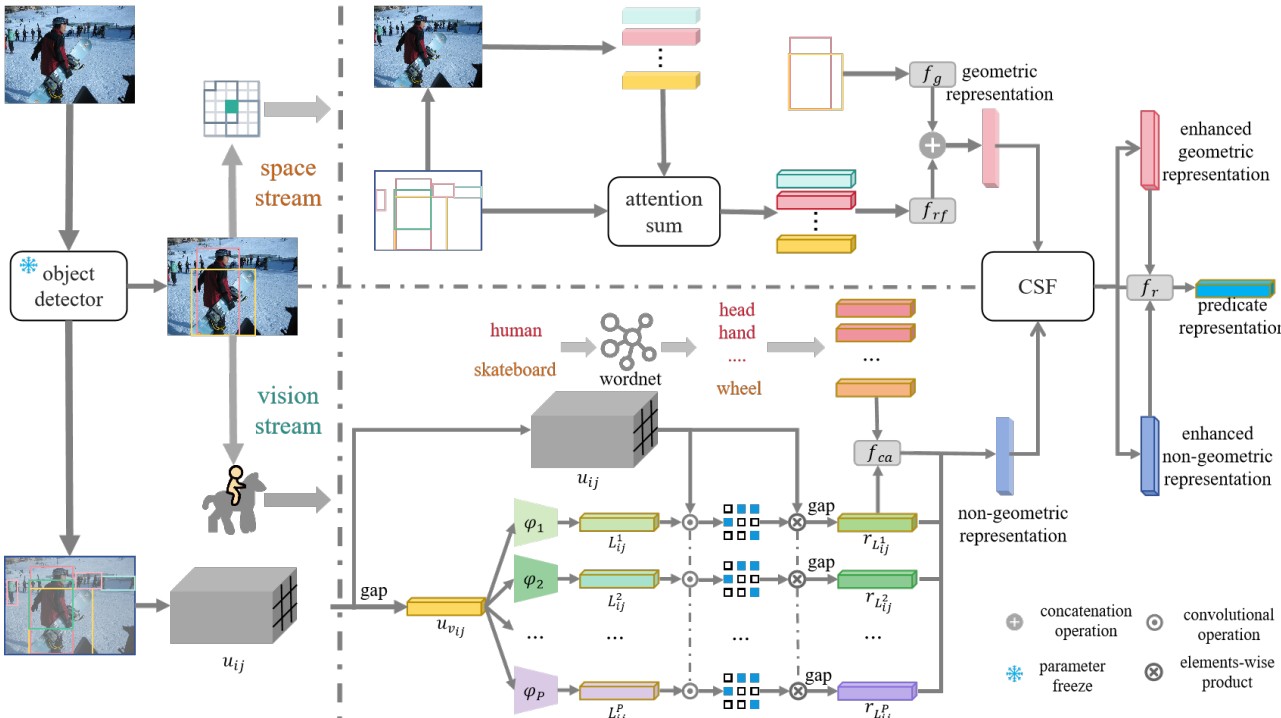

*Figure 2.* Illustration of overall pipeline of our DS-Net model. It contains two parallel streams: a **space stream** to predict geometric predicates from spatial layouts and edge features, and a **vision stream** to predict non-geometric predicates from fine-grained visual cues and linguistic priors. **CSF** denotes Cross-Stream Fusion module.

Thus, $L_{ij}^p$ is the $1 \times 1$ filter corresponding to the $p$-th latent part of the relationship proposal from object $i$ to $j$.

**Step 3: latent part feature extraction.** After latent part filter generation, each relationship proposal has $P$ convolutional filters $L_{ij} = \{L_{ij}^p\}_{p=1}^P$. Each generated filter $L_{ij}^p$ is then convolved with the original union feature map $u_{ij}$ to produce a spatial activation map $a_{ij}^p$:

$$a_{ij}^p = sigmoid(u_{ij} \odot L_{ij}^p), \tag{5}$$

where $\odot$ represents the convolutional operation. Each value in the spatial activation map represents how likely this local region contains the corresponding latent part. Then, we utilize these spatial activation maps as region-based attention weights to extract the corresponding latent parts $r_{L_{ij}} = \{r_{L_{ij}^p}\}_{p=1}^P$:

$$r_{L_{ij}^p} = gap(u_{ij} \otimes a_{ij}^p), \tag{6}$$

where $\otimes$ represents the elements-wise product. Through this step, we decouple the union feature used in the existing methods into $P$ latent parts to promote the model's attention to some fine-grained regions.

**Step 4: vision-semantic cross perception.** Finally, we combine these latent parts with the text of pre-defined parts to enhance their semantic meaning. Specifically, for each object category, we obtain a set of categories of its parts from an external knowledge source (such as, WordNet). For the $i$-th object proposal, if its detected class is non-part object, we can get a set of its parts $p_i = \{p_i^k\}_{k=1}^N$, representing $c_i$ has $N$ parts. And for the $i$-th object proposal, if its detected class is object part, its part is denoted as $p_i = c_i$.

We utilize Glove (Pennington et al., 2014) to encode these parts, and denote the text embedding of them as $p_{t_i} = \{p_{t_i}^k\}_{k=1}^N$. And then we use cross-attention strategy to refine the latent parts, with the latent parts as queries and the text embeddings as keys and values:

$$r_{v_{ij}} = sum(r_{L_{ij}} + f_{ca}(r_{L_{ij}}, p_{t_{ij}}, p_{t_{ij}})), \tag{7}$$

where $f_{ca}$ represents a cross attention strategy, $sum$ represents the sum of the features of P latent parts and $p_{t_{ij}} = \{p_{t_i}, p_{t_j}\}$. This step grounds the visually discovered parts in rich and semantic context.

### 3.4. Cross-Stream Fusion

After obtaining the geometric representation $r_{g_{ij}}$ from the space stream and the non-geometric representation $r_{v_{ij}}$ from the vision stream, the final step is to fuse them. Our CSF includes the following strategies:

**Vision-to-Space Fusion** ($CSF_{v2s}$). Firstly, we utilize non-geometric predicate representation to enhance geometric

predicate representation. Specifically, for the relationship proposal from object $i$ to $j$, we take its geometric representation $r_{g_{ij}}$ as query, its non-geometric representation $r_{v_{ij}}$ as key and value. And then, the enhanced geometric representation $r_{g_{ij}}^+$ can be computed by

$$r_{g_{ij}}^+ = r_{g_{ij}} + f_{ca}(r_{g_{ij}}, r_{v_{ij}}, r_{v_{ij}}), \tag{8}$$

**Space-to-Vision Fusion** ($CSF_{s2v}$). Similar to vision-to-space, for the relationship proposal from object $i$ to $j$, we take its non-geometric representation $r_{v_{ij}}$ as query, its geometric representation $r_{g_{ij}}$ as key and value. And then, the enhanced non-geometric representation $r_{v_{ij}}^+$ can be computed by

$$r_{v_{ij}}^+ = r_{v_{ij}} + f_{ca}(r_{v_{ij}}, r_{g_{ij}}, r_{g_{ij}}) + f_w(w_i \oplus w_j). \tag{9}$$

where $f_w$ is a fully-connected network that integrates the semantic features of objects.

Finally, the two mutually enhanced representations are concatenated to form the final predicate representation $r_{ij}$, which benefits from both specialized processing and collaborative fusion:

$$r_{ij} = f_r(r_{v_{ij}}^+ \oplus r_{g_{ij}}^+), \tag{10}$$

where $f_r$ is a fully-connected network to integrate the geometric representation and the non-geometric representation to represent predicate.

### 3.5. Learning Strategy

To train our DS-Net, we design a composite learning strategy. It contains the following three modules:

**Object Prediction.** To predicate the object, we introduce a linear classifier. It takes as input the object representation $o_i$, and the distribution of object $pred_{o_i}$ is computed as

$$pred_{o_i} = softmax(W_{obj}o_i), \tag{11}$$

where $W_{obj}$ is the parameter of object classifier.

**Auxiliary Module for Two Parallel Streams.** Then, we introduce two auxiliary modules to train our space stream and vision stream. Specifically, we introduce two additional linear classifiers. They utilize the representations extracted from space stream and vision stream as input, respectively, and use the ground-truth labels for training. For the relationship proposal from object $i$ to $j$, the distribution of geometric predicate $pred_{r_{g_{ij}}}$ is computed as

$$pred_{r_{g_{ij}}} = softmax(W_{ge}r_{g_{ij}}), \tag{12}$$

where $W_{ge}$ is the parameter of space stream classifier. And the distribution of non-geometric predicate $pred_{r_{v_{ij}}}$ is computed as

$$pred_{r_{v_{ij}}} = softmax(W_{ng}r_{v_{ij}}), \tag{13}$$

where $W_{ng}$ is the parameter of vision stream classifier. It is worth mentioning that these two auxiliary modules do not participate in the final predicate prediction. They only serve as an additional signal to constrain two parallel streams.

**Predicate Prediction.** Finally, we utilize a linear classifier to predict predicates. For the relationship proposal from object $i$ to $j$, our classifier integrates the predicate representation $r_{ij}$ and a class frequency prior $q_{ij}$ for classification, which is widely used in existing works (Zellers et al., 2018; Li et al., 2021; Zhang et al., 2024). The distribution of predicate $pred_{r_{ij}}$ is computed as

$$pred_{r_{ij}} = softmax(W_{rel}r_{ij} + q_{ij}), \tag{14}$$

where $W_{rel}$ is the parameter of predicate classifier.

**Training Loss.** To train our DS-Net model, we design a multi-tasks loss $L_{total}$ of four components, including $L_p$ for predicate classification, $L_o$ for object classification, $L_s$ and $L_v$ for auxiliary modules. Formally,

$$L_{total} = L_p + \lambda_o L_o + \lambda_s L_s + \lambda_v L_v \tag{15}$$

where $\lambda_o$, $\lambda_s$ and $\lambda_v$ are weight parameters for calibrating the supervision from each sub-task. Here, $L_p$, $L_o$, $L_s$, $L_v$ are the standard cross entropy loss for multi-class classification (foreground categories plus background).

## 4. Experiments

In general, to fully verify the effectiveness and versatility of our method, we conduct experiments on three standard SGG datasets, including Visual Genome (Krishna et al., 2017), Open Images V6 (Kuznetsova et al., 2020) and GQA (Hudson & Manning, 2019).

### 4.1. Experiments Configuration of Visual Genome

**Dataset Details.** Visual Genome (VG) consists of 108,073 images, including tens of thousands of unique object and predicate categories. In our experiments, we follow the most commonly used data splits proposed by Xu et al. (2017) and Zellers et al. (2018). The 150 most frequent object categories and the 50 most frequent predicate types are adopted for evaluation.

**Evaluation Protocol.** Following the most existing works (Zellers et al., 2018; Li et al., 2021; Hayder & He, 2024), we evaluate our model on three sub-tasks: 1) predicate classification (PredCls); 2) scene graph classification (SGCls); 3) scene graph generation (SGGen). In each task, we take **recall** (**R@K**), **mean recall** (**mR@K**) and **overall mean** (**M@K**) as evaluation metrics.

**Implementation Details.** In general, we introduce our implementation details from the following three aspects: **1) Object detector**: in our experiment, following the previous

*Table 1.* The performance of state-of-the-art SGG models on three SGG tasks with graph constraints setting on mR@50/100, R@50/100 and M@50/100 on the VG dataset. The **best** and the **second best** methods are marked according to formats.

| Models | PredCls | | | SGCls | | | SGGen | | |
|---|---|---|---|---|---|---|---|---|---|
| | mR@50/100 | R@50/100 | M@50/100 | mR@50/100 | R@50/100 | M@50/100 | mR@50/100 | R@50/100 | M@50/100 |
| VCTree (Tang et al., 2019) | 17.9/19.4 | **66.4/68.1** | 42.1/43.7 | 10.1/10.8 | 38.1/38.8 | 24.1/24.8 | 5.9/8.0 | 27.9/31.3 | 16.9/19.6 |
| Unbiased (Tang et al., 2020) | 25.4/28.7 | 47.2/51.6 | 36.3/40.1 | 12.2/14.0 | 25.4/27.9 | 18.7/20.9 | 9.3/11.1 | 19.4/23.2 | 14.3/17.1 |
| MSDN (Li et al., 2017) | 19.2/20.5 | 65.0/66.7 | 42.1/43.6 | 11.6/12.6 | 38.9/39.8 | 25.2/26.2 | 7.7/9.0 | 30.3/33.3 | 19.0/21.1 |
| GPS-Net (Lin et al., 2020) | 15.2/16.6 | 65.2/67.1 | 40.2/41.8 | 8.5/9.1 | 37.8/39.2 | 23.1/24.1 | 6.7/8.6 | **31.1/35.9** | 18.9/22.2 |
| SMN (Zellers et al., 2018) | 13.3/14.8 | 65.2/67.1 | 39.2/40.9 | 7.1/7.6 | 35.8/36.5 | 21.4/22.0 | 5.3/6.1 | 27.2/30.3 | 16.2/18.2 |
| BGNN (Li et al., 2021) | 30.4/32.9 | 59.2/61.3 | 44.8/47.1 | 14.3/16.5 | 37.4/38.5 | 25.8/27.5 | 10.7/12.6 | 31.0/35.8 | 20.8/24.2 |
| Dual-ResGCN (Zhang et al., 2020) | 19.7/21.5 | 66.6/68.2 | 43.1/44.8 | 11.1/12.0 | 38.3/39.1 | 24.7/25.5 | 8.4/9.5 | 28.1/31.5 | 18.2/20.5 |
| PPDL (Li et al., 2022b) | 32.2/33.3 | 47.2/47.6 | 39.7/40.4 | 17.5/18.2 | 28.4/29.3 | 22.9/23.7 | 11.4/13.5 | 21.2/23.9 | 16.3/18.7 |
| Nice-Motif (Li et al., 2022a) | 29.9/32.3 | 55.1/57.2 | 42.5/44.7 | 16.6/17.9 | 33.1/34.0 | 24.8/25.9 | 12.2/14.4 | 27.8/31.8 | 20.0/23.1 |
| HetSGG (Yoon et al., 2023) | 31.6/33.5 | 57.8/58.9 | 44.7/46.2 | 17.2/18.7 | 37.6/38.5 | 27.4/28.6 | 12.2/14.4 | 30.0/34.6 | 21.1/24.5 |
| EdgeSGG (Kim et al., 2023) | **34.7/36.9** | 60.1/61.8 | **47.4/49.3** | **17.8/18.8** | **39.1/40.1** | **28.4/29.4** | **13.6/15.8** | 29.7/34.0 | **21.6/24.9** |
| ST-SGG (Kim et al., 2024) | 28.1/31.5 | 53.9/57.7 | 41.0/44.6 | 16.9/18.0 | 33.4/34.9 | 25.1/26.4 | 11.6/14.2 | 26.7/30.7 | 19.1/22.4 |
| DS-Net | **40.2/41.1** | 62.4/63.9 | **51.3/52.5** | **21.2/22.4** | **41.2/42.2** | **31.2/31.8** | **18.1/18.7** | 34.7/36.9 | **26.4/27.8** |

*Table 2.* The performance of our method combined with other advanced methods on three SGG tasks with graph constraints setting on the VG dataset. The **increased** values, the **max increased** values and the **min increased** values are marked according to formats.

| Models | PredCls | | | SGCls | | | SGGen | | |
|---|---|---|---|---|---|---|---|---|---|
| | mR@50/100 | R@50/100 | M@50/100 | mR@50/100 | R@50/100 | M@50/100 | mR@50/100 | R@50/100 | M@50/100 |
| BGNN (Li et al., 2021) | 30.4/32.9 | 59.2/61.3 | 44.8/47.1 | 14.3/16.5 | 37.4/38.5 | 25.8/27.5 | 10.7/12.6 | 31.0/35.8 | 20.8/24.2 |
| DS-Net | 40.2/41.1 (**8.2**) | 62.4/63.9 (**2.6**) | 51.3/52.5 (**5.4**) | 21.6/22.4 (**5.9**) | 40.8/41.6 (**3.1**) | 31.2/31.8 (**4.3**) | 18.1/18.7 (**6.1**) | 34.7/36.9 (**1.1**) | 26.4/27.8 (**3.6**) |
| PENET (Zheng et al., 2023b) | 31.5/33.8 | 68.2/70.1 | 49.8/51.9 | 17.8/18.9 | 39.4/40.7 | 28.6/29.8 | 12.4/14.5 | 30.7/35.2 | 21.5/24.8 |
| PENET + DS-Net | 39.6/40.7 (**6.9**) | 69.9/72.2 (**2.1**) | 52.3/54.8 (**2.9**) | 23.6/24.2 (**4.7**) | 41.2/42.6 (**1.6**) | 32.4/33.4 (**3.6**) | 18.4/19.6 (**5.1**) | 35.4/36.2 (**1.0**) | 26.9/27.9 (**3.1**) |
| DRM (Li et al., 2024a) | 47.1/49.6 | 43.9/45.8 | 45.5/47.7 | 27.8/29.2 | 27.5/28.4 | 27.6/28.8 | 20.4/24.1 | 19.0/22.9 | 19.7/23.5 |
| DRM + DS-Net | 49.4/51.9 (**2.3**) | 44.6/46.9 (**1.1**) | 46.1/49.2 (**1.5**) | 29.6/30.7 (**1.5**) | 29.9/30.8 (**2.4**) | 29.7/30.9 (**2.1**) | 23.9/25.5 (**1.4**) | 21.9/24.1 (**1.1**) | 22.0/25.0 (**1.5**) |
| RepSGG (Liu & Bhanu, 2024) | 39.7/43.7 | 27.8/28.8 | 33.7/36.2 | 22.3/27.7 | 17.9/20.3 | 20.1/24.0 | 15.3/18.9 | 12.1/14.6 | 13.7/16.7 |
| RepSGG + DS-Net | 44.2/47.8 (**4.1**) | 30.3/31.0 (**2.2**) | 37.2/39.4 (**3.2**) | 27.1/30.9 (**3.2**) | 20.9/22.2 (**1.9**) | 24.0/26.6 (**2.6**) | 18.4/21.7 (**2.4**) | 15.5/16.3 (**1.7**) | 17.0/19.0 (**2.3**) |
| RA-SGG (Yoon et al., 2024) | 36.2/39.1 | 62.2/64.1 | 49.2/51.6 | 20.9/22.5 | 38.2/39.1 | 29.5/30.8 | 14.1/17.1 | 26.0/30.3 | 20.2/23.7 |
| RA-SGG + DS-Net | 43.2/45.8 (**6.7**) | 63.9/65.4 (**1.3**) | 53.5/55.6 (**4.0**) | 23.4/24.9 (**2.4**) | 38.9/40.0 (**0.9**) | 31.1/32.5 (**1.7**) | 18.2/20.7 (**3.6**) | 27.9/32.2 (**1.9**) | 23.1/26.5 (**2.8**) |
| CAModule (Liu et al., 2025) | 36.7/39.3 | 59.8/63.4 | 48.2/51.3 | 21.1/24.7 | 36.8/38.2 | 28.9/31.4 | 16.3/18.2 | 29.1/32.7 | 22.7/25.4 |
| CAModule + DS-Net | 41.1/42.4 (**3.1**) | 62.3/64.7 (**1.3**) | 51.7/53.5 (**2.2**) | 24.5/26.4 (**1.7**) | 38.9/40.2 (**2.0**) | 31.7/33.3 (**1.9**) | 19.1/19.7 (**1.5**) | 31.8/34.3 (**1.6**) | 25.4/27.0 (**1.6**) |
| RcSGG (Sun et al., 2025) | 38.8/41.4 | 54.3/57.3 | 46.5/49.3 | 23.2/24.1 | 32.8/33.1 | 28.0/28.6 | 16.9/19.9 | 24.8/27.9 | 20.8/23.9 |
| RcSGG + DS-Net | 42.4/43.6 (**2.2**) | 56.2/58.1 (**0.8**) | 49.3/50.8 (**1.5**) | 24.7/25.6 (**1.5**) | 34.5/36.2 (**3.1**) | 29.6/30.9 (**2.3**) | 18.9/21.2 (**1.3**) | 26.8/30.1 (**2.2**) | 22.8/25.6 (**1.7**) |

works (Li et al., 2021; 2024a), we adopt the pre-trained Faster-RCNN with ResNeXt-101-RPN (Xie et al., 2017) as object detector to obtain the object and relationships proposals; **2) Parameter setting**: our experiments were performed on four 3090 GPUs. The batch size and initial learning rate are set to 12 and 0.024, respectively. And $\lambda_o$, $\lambda_s$, $\lambda_v$ in Eq.15 are set to 1, 0.5 and 0.5, respectively. Meanwhile, our model is optimized by the Adam algorithm with the momentum of 0.9 and 0.999. **3) The number of latent parts** $P$: Experiments show that the best performance is achieved when $P$ is set to **8**.

### 4.2. Comparisons with State-of-the-Art Methods

According to the research content of the two-stage scene graph generation methods in recent years, the SOTA methods we compared are mainly divided into two aspects:

**1) Effectiveness on balanced predicate recognition.** We first compare DS-Net with recent SGG methods that enhance predicate representations with contextual cues, under the same evaluation protocol. Results are summarized in Table 1. **(i) Mean recall.** DS-Net yields consistent gains in

mR@K across all three tasks. In particular, it improves over EdgeSGG by **4.2**%, **3.6**%, and **2.9**% in mR@100 on PredCls, SGCls, and SGGen, respectively. As we mentioned in Section 1, non-geometric predicates are typically scarce in SGG datasets. Thus, these results demonstrate that our method effectively enhances the model's capability to recognize rare non-geometric predicates. **(ii) Recall.** It is worth mentioning that our DS-Net is based on Li et al. (2021) (shadow background in Table 1). Consequently, our R@K performance remains competitive. This suggests that the overall recall on frequent geometric predicates is preserved. Overall, these results validate that our method effectively strikes a balance between geometric and non-geometric predicates, which aligns well with our primary objective.

**2) Generalization of our method.** We then evaluate whether DS-Net generalizes across different SGG methods by plugging it into several representative methods and keeping the remaining components unchanged. Results are summarized in Table 2. Across all tested backbones, DS-Net consistently improves mR@K on PredCls/SGCls/SGGen, indicating better balanced recognition beyond a single architecture. On the challenging SGGen setting, DS-Net brings

*Table 3.* The performance of our ablation study on different settings on three SGG tasks with graph constraints setting on the VG dataset. The complete model is shown in the shadow row in the table.

| Settings | Space Stream | | Vision Stream | | Cross-Stream Fusion | | | PredCls | | SGCls | | SGGen | |
|---|---|---|---|---|---|---|---|---|---|---|---|---|---|
| | $S_{ours}$ | $S_{common}$ | $V_{ours}$ | $V_{common}$ | $CSF_{concate}$ | $CSF_{v2s}$ | $CSF_{s2v}$ | mR@50/100 | R@50/100 | mR@50/100 | R@50/100 | mR@50/100 | R@50/100 |
| | - | - | - | - | - | - | - | 30.4/32.9 | 59.2/61.3 | 14.3/16.5 | 37.4/38.5 | 10.7/12.6 | 31.0/35.8 |
| a | ✓ | ✗ | ✓ | ✗ | ✗ | ✓ | ✓ | **40.2/41.1** | **62.4/63.9** | **21.2/22.4** | **41.2/42.2** | **18.1/18.7** | **34.7/36.9** |
| b | ✗ | ✓ | ✓ | ✗ | ✗ | ✓ | ✓ | 38.1/39.3 | 61.7/63.2 | 20.3/21.6 | 40.6/40.9 | 17.6/18.1 | 33.1/36.2 |
| c | ✓ | ✗ | ✗ | ✓ | ✗ | ✓ | ✓ | 36.2/38.4 | 61.9/63.4 | 19.4/19.9 | 40.7/41.3 | 17.2/18.6 | 33.5/34.9 |
| d | ✓ | ✗ | ✓ | ✗ | ✓ | ✗ | ✗ | 37.5/39.1 | 61.1/63.5 | 19.7/20.9 | 39.9/41.2 | 16.9/18.2 | 34.6/35.7 |
| e | ✓ | ✗ | ✗ | ✗ | ✗ | ✗ | ✗ | 32.5/34.7 | 59.9/61.7 | 15.9/17.4 | 37.8/38.9 | 12.4/13.3 | 31.6/36.3 |
| f | ✗ | ✗ | ✓ | ✗ | ✗ | ✗ | ✗ | 34.9/36.1 | 57.4/59.5 | 17.4/18.6 | 37.1/38.3 | 14.4/15.6 | 25.6/29.3 |
| g | ✓ | ✗ | ✓ | ✗ | ✗ | ✓ | ✗ | 38.4/40.3 | 62.2/63.4 | 20.4/21.3 | 40.3/41.4 | 17.2/17.9 | 34.3/36.1 |
| h | ✓ | ✗ | ✓ | ✗ | ✗ | ✗ | ✓ | 39.1/40.9 | 61.4/62.6 | 20.8/21.7 | 39.9/40.7 | 17.6/18.2 | 33.9/36.0 |

*Table 4.* The performance of our ablation study on different numbers of latent parts on two SGG tasks. The **best** method is marked according to formats.

| Numbers | PredCls | | SGCls | |
|---|---|---|---|---|
| | mR@50/100 | R@50/100 | mR@50/100 | R@50/100 |
| 4 | 37.4/38.9 | 61.2/62.4 | 19.4/20.3 | 39.4/40.4 |
| 6 | 37.9/39.2 | 62.1/63.4 | 19.6/20.9 | 40.5/41.1 |
| 8 | **40.2/41.1** | **62.4/63.9** | **21.2/22.4** | **41.2/42.2** |
| 10 | 39.9/40.7 | 62.1/63.4 | 20.6/21.9 | 40.8/41.8 |
| 12 | 40.0/40.9 | 61.9/63.1 | 20.9/22.0 | 40.7/41.6 |
| 14 | 39.9/41.0 | 62.0/63.7 | 20.3/21.8 | 40.9/41.4 |

*Table 5.* Performance comparison with the SOTA methods on Open Images V6 dataset. The **best** and the **second best** methods are marked according to formats.

| Method | mR@50 | R@50 | $wmAP_{rel}$ | $wmAP_{phr}$ | $score_{wtd}$ |
|---|---|---|---|---|---|
| VCTree (Tang et al., 2019) | 33.9 | 74.1 | 34.2 | 33.1 | 40.2 |
| RelDN (Lin et al., 2020) | 37.2 | 75.3 | 32.2 | 33.4 | 42.0 |
| Motifs (Zellers et al., 2018) | 32.7 | 71.6 | 29.9 | 31.6 | 38.9 |
| BGNN (Li et al., 2021) | 40.5 | 75.0 | 33.5 | 34.1 | 42.1 |
| HetSGG (Yoon et al., 2023) | 42.7 | 76.8 | 34.6 | 35.5 | 43.3 |
| Unbiased (Tang et al., 2020) | 35.5 | 69.3 | 30.7 | 32.8 | 39.3 |
| PENET (Zheng et al., 2023b) | - | 76.5 | 36.6 | 37.4 | 44.9 |
| EdgeSGG (Kim et al., 2023) | **43.3** | **77.1** | **36.4** | **37.4** | **44.9** |
| IWSL (Liu et al., 2023) | 42.2 | 74.7 | 33.1 | 34.3 | 41.9 |
| DS-Net | **46.4** | **77.5** | **38.2** | **39.1** | **46.4** |

**1.3% ∼ 5.1%** absolute gains in mR@100 and **1.0% ∼ 2.2%** gains in R@100, showing that the improvements are not limited to tail-focused metrics. These results demonstrate that our strategy can be effectively integrated into other existing methods, which aligns with the claims made in Section 1. More importantly, these results highlight the importance of synergistic modeling of geometric and non-geometric predicates for generating semantically richer scene graphs.

**Running efficiency analysis.** For the two-stage SGG methods, running efficiency is not the primary concern. Thus, we provide it here as a reference. Using BGNN as a baseline, the running efficiency of BGNN on a single V100 is **1.7 FPS**, while our DS-Net achieves a competitive **1.5 FPS**.

## 4.3. Experiments of Open Images V6

**Dataset Details.** Open Images V6 dataset (Kuznetsova et al., 2020) is a large-scale dataset commonly used for SGG tasks. It contains a diverse collection of over 133k images with 126,368 training, 1,813 validation, and 5,322 testing images. This dataset provides object-level annotations for each image, including bounding boxes and 301 object categories. In addition, it includes 31 relationship annotations that describe the interactions between pairs of objects within a scene.

**Metrics.** For this dataset, we follow the same data processing and evaluation protocols in the existing works (Li et al., 2021; Zhang et al., 2019b). The mR@50, R@50, weighted

mean AP of relationships ($wmAP_{rel}$), and weighted mean AP of phrase ($wmAP_{phr}$) are used as evaluation metrics. Following standard evaluation metrics of Open Images refers to (Li et al., 2021; Zhang et al., 2019b), the weight metric $score_{wtd}$ is computed as: $score_{wtd} = 0.2 \times R@50 + 0.4 \times wmAP_{rel} + 0.4 \times wmAP_{phr}$.

**Quantitative Results.** The quantitative results are shown in Tab.5. Our method achieves the SOTA performance on mean recall and competitive results on weighted metric score. The excellent performance of all metrics fully verifies the effectiveness and generalization of our method.

## 4.4. Experiments of GQA

**Dataset Details.** GQA (Hudson & Manning, 2019) is a large-scale visual question answering dataset with images for the VG dataset and balanced question-answer pairs. Following (Dong et al., 2022; Liu et al., 2025), we select the top-200 object classes and top-100 relationship classes, adhering to its split of 70% for training (including a 5k validation set) and 30% for testing.

**Quantitative Results.** The quantitative results are shown in Tab.6. It is worth mentioning that: to ensure a fair comparison, all methods in Tab.6 use Motif (Zellers et al., 2018) as the backbone for the experiments. Our method achieves competitive or SOTA performance on all benchmarks. The excellent performance of all metrics fully verifies the effec-

*Table 6.* Performance comparison with the SOTA methods on GQA dataset. The **best** method is marked according to formats.

| Methods | PredCls | | SGCls | | SGGen | |
|---|---|---|---|---|---|---|
| | mR@50 | mR@100 | mR@50 | mR@100 | mR@50 | mR@100 |
| Motifs (Zellers et al., 2018) | 15.4 | 16.3 | 8.6 | 9.4 | 6.8 | 8.6 |
| GCL (Dong et al., 2022) | 36.7 | 38.1 | 17.3 | 18.1 | 16.8 | **18.8** |
| CFA (Li et al., 2023) | 31.7 | 33.8 | 14.2 | 15.2 | 11.6 | 13.2 |
| EICR (Min et al., 2023) | 36.3 | 38.0 | 17.2 | 18.2 | 16.0 | 18.0 |
| RcSGG (Sun et al., 2025) | 36.9 | 38.5 | 16.7 | 18.4 | 15.8 | 17.6 |
| DS-Net | **37.3** | **38.9** | **19.4** | **20.9** | **16.9** | 18.6 |

*Table 7.* The performance of mR@100 on different predicate super-types under PredCls on the VG dataset.

| Predicate Type | classes | Instances | BGNN | DS-Net |
|---|---|---|---|---|
| Geometric | 15 | 228k | 29.1 | **37.2** |
| Semantic | 24 | 39k | 32.1 | **43.9** |
| Possessive | 11 | 188k | 28.6 | **36.1** |

tiveness and generalization of our method.

## 4.5. Ablation Study

The symbols in Table 3 are briefly introduced as follows: 1) $S_{our}$, $V_{ours}$, $CSF_{v2s}$ and $CSF_{s2v}$ represent the methods proposed by us; 2) $S_{common}$, $V_{common}$ represent the methods used in the existing works; 3) $CSF_{concate}$ represents directly concatenating the representations of two streams. More details can be found in our appendix. Based on them, our ablation study includes the following four aspects:

**Sensitivity to the number of latent parts $P$.** As shown in Table 4, experiments show that the best performance is achieved when $P$ is set to **8**, and the model's performance does not change significantly with variations in $P$. Meanwhile, we visualize the spatial activation maps of the latent parts for the "girl" and "bat" in Figure 3. As observed, the model pays attention to specific regions of the girl (e.g., "head", "hands") and the bat (e.g., "handle", "face"). It proves that our vision stream can focus on some details.

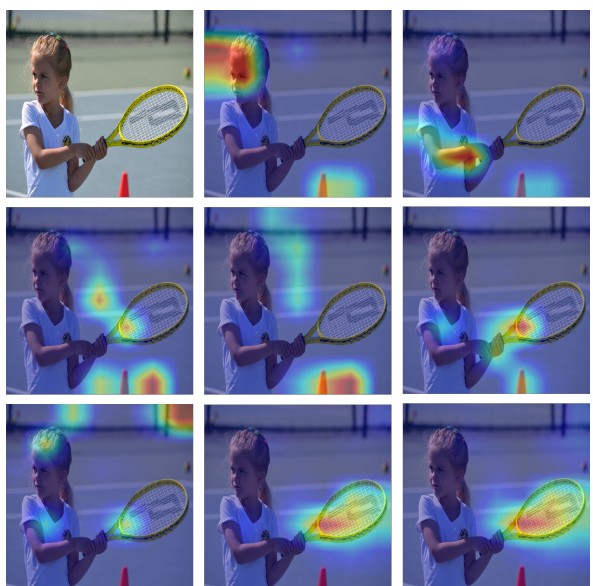

*Figure 3.* The spatial activation maps visualizing the latent parts $P$ ($P$ = 8) of the "girl" and the "bat".

**Necessity of stream specialization.** Using only a single stream substantially degrades performance compared to the full dual-stream model (as shown in Table 3: e and f). This indicates that geometric and non-geometric predicates require complementary cues, and neither stream alone is sufficient for strong predicate inference.

**Contribution of each components.** Replacing the baseline representation with our space-stream design consistently improves results, and similarly for the vision-stream design (as shown in Table 3: b and c). Moreover, cross-stream fusion further boosts performance over simple concatenation, suggesting that the gains are not merely from increasing feature capacity (as shown in Table 3: d). We also observe that enabling fusion in either direction improves performance, while bidirectional fusion yields the strongest results, supporting our hypothesis that the two predicate types can act as mutual priors (as shown in Table 3: g and h).

**Performance on each predicate super-type.** We further demonstrate the performance of our method on each super-type of the VG dataset. All results shown in Table 7 follow the same training configuration, and the split of super-types strictly follows the setting of Zellers et al. (2018). These results prove that our method can significantly improve the performance of each predicate super-type.

## 5. Conclusion

In this paper, we revisit the long-standing bias toward geometric predicates in existing SGG methods. Beyond the commonly cited long-tailed data distribution, we identify co-occurrence-induced representation entanglement as an additional factor that hinders balanced predicate recognition, where heterogeneous predicates that frequently co-occur are encoded into overly similar representations. To address this problem, we introduce DS-Net, a dual-stream synergistic method that explicitly models geometric and non-geometric predicates with specialized representations while preserving their mutual dependencies through cross-stream fusion. Extensive experiments on multiple benchmarks demonstrate that DS-Net consistently improves balanced predicate recognition across different SGG settings and can be seamlessly integrated into existing frameworks. We believe this work highlights the importance of explicitly modeling predicate heterogeneity and co-occurrence structure in SGG, and provides a practical direction for improving semantic richness in structured visual understanding.

## Impact Statement

This paper presents work whose goal is to advance the field of Machine Learning. There are many potential societal consequences of our work, none which we feel must be specifically highlighted here.

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

To better illustrate the main content of our idea, our appendix contains the following four aspects: 1) The problem setting and related works (in Sec.A); 2) The detailed explanations of our method (in Sec.B); 3) More detailed experimental analysis (in Sec.C);

# A. Problem Setting and Related Works

## A.1. Problem Setting and Method Overview

**Problem Setting.** Given an image $\mathbf{I}$, scene graph generation (SGG) aims to parse the input $\mathbf{I}$ into a scene graph $G = \{E, R\}$, where $E$ denotes the set of nodes representing objects, and $R$ represents the set of edges that encode the predicates between ordered pairs of entities. Typically, each node $e_i \in E$ is assigned a category label from a pre-defined set of object class $C_e$ and is associated with a corresponding image location indicated by a bounding box. Additionally, each edge $r_{ij} \in R$, which connects a pair of nodes $e_i$ and $e_j$, is linked to a predicate label derived from a specified set of predicate classes $C_p$ relevant to this task.

**Method Overview.** Scene graph generation (SGG), which parses an image into a structured graph of objects and their predicates, has received extensive attention (Li et al., 2024b; Zhao et al., 2024; Lin et al., 2024). Most existing SGG models adopt a two-stage paradigm that first detects objects and then infers pairwise predicates (Krishna et al., 2017; Tang et al., 2019; Chen et al., 2019; Zhang et al., 2019b; Lu et al., 2021). They are mostly dedicated to generating predicate representations for the detected objects, based on their visual appearance, relative positions, and contextual cues (Lu et al., 2016; Li et al., 2021; 2024a). However, due to the predicate representation ambiguity arising from spatial co-occurrence, the generated scene graphs are often factually correct, but semantically shallow. To address this problem, we propose a novel method, Dual-stream Synergistic Network (DS-Net), which infers predicates through the collaborative processing of space and vision.

Technically, our DS-Net contains two streams: a space stream and a vision stream. Among them, the space stream focuses on the spatial layout and edge features of objects to predict geometric predicates, while the vision stream infers non-geometric predicates through some fine-grained visual features and linguistic priors of objects. Through these two parallel streams, we extract the geometric and non-geometric representations for each pair of objects. Based on them, we then design Cross-Stream Fusion (CSF) module to enhance the corresponding predicate representation by using the mutual information of the two types. It utilizes the geometric representation as a spatial prior to enhance non-geometric prediction, and uses the rich visual cues in the non-geometric representation to enhance geometric interpretation. Through these strategies, our DS-Net no longer treats the two predicate types as the conflicting signals that need to be disentangled. Instead, it utilizes their synergy to facilitate predicate inference, providing a new perspective on resolving predicate ambiguity.

## A.2. Related Works

**Studies of Visual Relationships.** Spanning the past and the present, computer vision research is dedicated to understanding the content of visual scenes. As an important part of the visual scene, the visual relationship, which describes the interaction between subject and object, also receives extensive attention, such as in visual question answering (Qian et al., 2024; Lin et al., 2024; Gao et al., 2024), visual relationship detection (Li et al., 2024c; Lu et al., 2016; Liang et al., 2018), and scene graph generation (Li et al., 2024b; Zhao et al., 2024; Lin et al., 2024).

Early explorations of visual relationships tend to classify each "subject-predicate-object" triple as a single and holistic category (Divvala et al., 2014; Ramanathan et al., 2015). However, this approach quickly encounters a significant bottleneck: combinatorial explosion. As the number of object and predicate categories grows, the number of possible unique triplets skyrockets, leading to an unmanageable number of output classes. It creates a severe long-tail distribution problem, where a few common relationships (e.g., person-on-chair) dominate the training data, while the vast majority of meaningful interactions are too sparsely represented for the model to learn effectively.

To mitigate this scalability issue, the research community shifts towards the strategies that decouple the holistic category, learning the subject, object, and predicate components independently (Krishna et al., 2017; Tang et al., 2019; Chen et al., 2019; Zhang et al., 2019b). In this progress, Lu et al. (2016) make a key contribution. They propose a novel method that combines language priors to guide visual relationship detection (VRD). It first employs an object detector to identify candidate objects and their visual features. Subsequently, a language model is used to predict the predicate by evaluating the statistical likelihood of a given triplet. Based on this method, the existing two-stage research method paradigm basically forms: a model first detects objects and then infers pairwise predicates (Li et al., 2021; 2022a;b; Tang et al., 2020; Zellers

et al., 2018; Hayder & He, 2024).

In recent years, a primary focus of SGG research is the design of more sophisticated message passing mechanisms to more effectively capture and leverage scene context. By treating the scene as a graph, these methods utilize strategies based on Graph Neural Networks (GNNs) to refine node (object) and edge (relationship) representations (Xu et al., 2017; Dai et al., 2017; Li et al., 2017; Woo et al., 2018; Wang et al., 2019a). Some methods use sequential models (LSTMs) or fully-connected graphs to model context, allowing every object to influence every other. Other work explores more structured and sparse graph structures, pruning the fully-connected graph based on geometric or semantic heuristics to reduce noise and computational overhead (Tang et al., 2019; Yin et al., 2018). However, these predefined graph structures often lack flexibility. To overcome this rigidity, new research introduces adaptive messaging strategies (Li et al., 2021; Kim et al., 2023; 2024). These models can dynamically learn the weights of message-passing routes, enabling the network to calculate more flexible and relevant contextual representations for each relationship prediction.

**Studies of Geometric Predicate.** Geometric predicates, which describe the spatial interactions between objects (Zheng et al., 2019), are fundamentally contingent upon the three-dimensional positions of those objects in a scene. Accurately inferring these predicates from a single 2D image presents a significant and long-standing challenge in computer vision. The core difficulty arises from the inherent ambiguity of a 2D projection: while standard object detectors can provide precise planar coordinates via bounding boxes, they offer insufficient information to resolve depth, scale, and relative positioning along the viewing axis. Thus, relying solely on 2D detections makes it nearly impossible to distinguish between complex spatial arrangements.

To overcome this limitation, our work draws inspiration from the extensive body of research in single-view 3D scene reconstruction (Roberts, 1963; Hedau et al., 2009; Lee et al., 2009; Dasgupta et al., 2016). Perceiving and reconstructing 3D spatial information from a flat image is a central goal of the field. The common strategy to make this ill-posed problem more tractable is to simplify the task by incorporating supplementary data or priors. Many existing studies leverage the pre-trained depth estimation modules to generate pseudo-3D information (Ma et al., 2020; Wang et al., 2019b; Weng & Kitani, 2019), rely on manually annotated 3D key-points to ground their models (Barabanau et al., 2019), or utilize extensive libraries of 3D CAD models for alignment and comparison (Manhardt et al., 2019).

Within this domain, a key focus is developing methods to estimate an object's 3D properties directly from 2D image cues. For instance, Chen et al. (2016) introduce a method that samples 3D bounding box candidates by assuming a ground plane, then scores these proposals using a rich combination of semantic segmentation, object shape priors, and contextual information. Similarly, Ku et al. (2019) estimate per-instance point clouds and enforce alignment between the projected point cloud and the object's appearance in the image to refine their 3D proposals. In another approach, Liu et al. (2019) generate a multitude of 3D proposals and select the best one by measuring the fitting degree between the projected 3D box and the 2D detection. A central theme across these works is the effort to effectively leverage available 2D information to infer latent 3D positions.

Inspired by the holistic approach in Total3D Understanding, we propose a method that, instead of relying on external data, focuses on synergistically combining intrinsic 2D cues (Nie et al., 2020). Specifically, we comprehensively consider both the 2D bounding box coordinates and the visual appearance features of an object. By integrating these information, we aim to generate an enriched spatial feature representation that implicitly encodes 3D properties, thereby enabling a more accurate prediction of the geometric predicates between objects.

## B. Detailed explanations of our method

We introduce a new architecture called the Dual-stream Synergistic Network (DS-Net), which aims to solve a core challenge in Scene Graph Generation (SGG): predicate representation ambiguity arising from spatial co-occurrence. Our core insight is that acknowledging the coexistence of geometric and non-geometric predicates, rather than struggling to disentangle them, is better suited for predicate inference than existing single-stream architectures (Xu et al., 2017; Lu et al., 2021; Li et al., 2021; Kim et al., 2024; Yoon et al., 2024). Many existing methods struggle to distinguish between a semantically rich, non-geometric relationship (e.g., "person-riding-horse") and a visually concurrent, more common but semantically shallow geometric relationship (e.g., "person-on-horse"). Because datasets are filled with these simple geometric relationships (Zellers et al., 2018), models develop a bias, leading to generated scene graphs that are factually correct but lack semantic depth. Our core contributions can be summarized in the following three aspects:

**1) We provide a novel perspective to solve predicate representation ambiguity.** First, we are the first to systematically

*Table 8.* Predicate division criteria in VG dataset.

| Type | Examples | | | | | Classes | Instances |
|---|---|---|---|---|---|---|---|
| | All Predicates in VG dataset | | | | | | |
| Geometric | above
at
in front of | across
on back of
near | against
behind
on | along
between
over | and
in
under | 15 | 228k (50%) |
| Non-Geometric | attached to
eating
hanging from
lying on
parked on
sitting on
walking on | belonging to
flying in
has
made of
part of
standing on
watching | carry
for
holding
mounted on
playing
to
wearing | covered in
from
laying on
of
riding
using
wears | covering
growing on
looking at
painted on
says
walking in
with | 35 | 227k (50%) |

analyze the problem of predicate representation ambiguity from the perspective of "spatial co-occurrence", and we propose a disruptive core concept: a shift from "disentanglement" to "synergy". Previous researches have largely focused on "disentangling" or "separating" geometric relationships from non-geometric ones within mixed visual features. However, due to the spatial co-occurrence of geometric and non-geometric predicates, it is unrealistic to completely decouple them. However, we argue that these two types of relationships are not conflicting signals but rather co-existing and complementary ones. Our core insight is that acknowledging the co-existence of geometric and non-geometric predicates, rather than struggling to disentangle them, is better suited for predicate inference. We believe that geometric relationships can provide a powerful spatial prior for non-geometric ones, and vice versa. This conceptual shift is the cornerstone of our work and provides a novel perspective for resolving predicate ambiguity.

**2) We introduce a novel method, DS-Net.** It infers predicates through collaborative processing of space and vision. Among them, the space stream focuses on the spatial layout and edge features of objects to predict geometric predicates, while the vision stream infers non-geometric predicates through some fine-grained visual features and linguistic priors of objects. Through these two parallel streams, we extract the geometric and non-geometric representations for each pair of objects. Based on them, we then design CSF module to enhance the corresponding predicate representation by using the mutual information of the two types. It uses geometric representation as a spatial prior to enhance non-geometric prediction, and uses the rich visual cues in non-geometric representation to enhance geometric interpretation. By the collaborative processing of these streams, our DS-Net no longer treats the two predicate types as conflicting signals that need to be disentangled. Instead, it utilizes their synergy to facilitate predicate inference, providing a new perspective on resolving predicate ambiguity.

**3) We have proved the effectiveness and versatility of our method through a large number of experiments.** Specifically, our DS-Net achieves significant improvements in various metrics, especially in the **mR@K** metric that better reflects unbiased performance. More importantly, our method is highly versatile and can be efficiently integrated into a variety of the existing SGG models and significantly improve their performance.

## C. Additional Experimental Analysis

In the main paper, we have provided the detailed quantitative analysis of the overall model and its individual modules. In this section, we present the additional quantitative analysis as well as qualitative results.

### C.1. Descriptions of Some Symbols

In general, the symbols in Tab.3 contain the following three parts:

**Space Stream:** This part includes our improvement for the geometric predicates. In Tab.3, $S_{ours}$ represents the method mentioned in our paper (Eq.2). And $S_{common}$ represents using existing methods to extract the geometric representation, which means that Eq.2 is modified to:

$$r_{g_{ij}} = f_{rb}(u_{ij} \oplus g_i \oplus g_j), \tag{16}$$

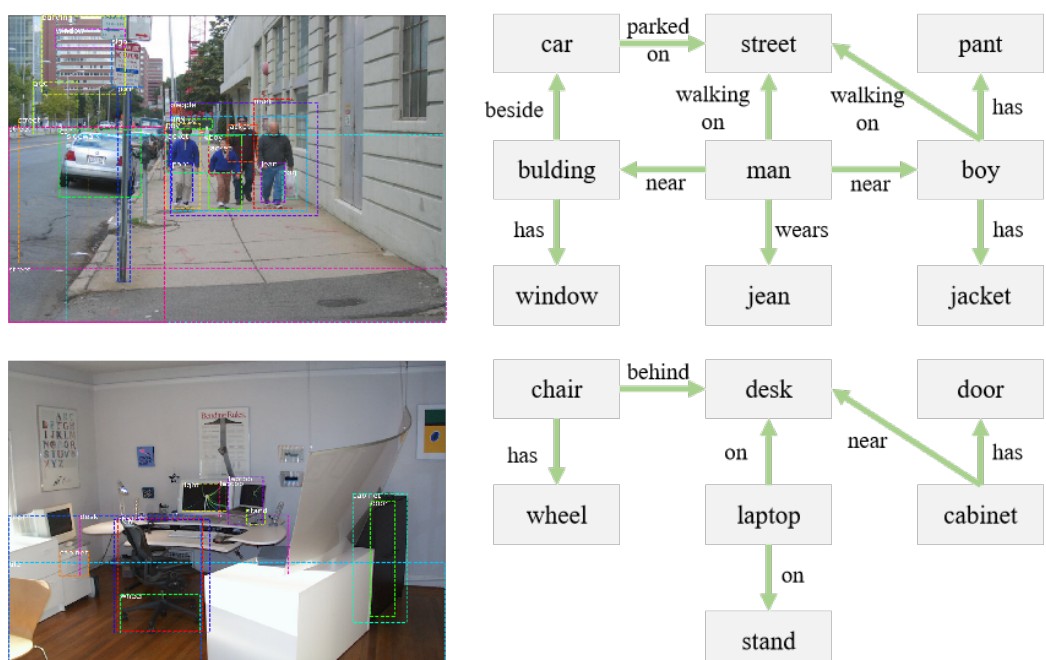

*Figure 4.* The visualized results of our method.

where $f_{rf}$ is a projection MLP.

**Vision Stream:** This part includes our improvement for the non-geometric predicates. In Tab.3, $V_{ours}$ represents the method mentioned in our paper (Eq.7). And $V_{common}$ represents using existing methods to extract the geometric representation, which means that Eq.7 is modified to Eq.16.

**Cross-Stream Fusion.** In Tab.3, this part contains three symbols. Among them, $CSF_{v2s}$ and $CSF_{s2v}$ are consistent with the definitions in Sec.3.4. And $CSF_{concate}$ represents removing CSF module and directly concatenating the two representations. These settings are designed to verify the effectiveness of our proposed method.

### C.2. Detailed Predicate Division

In general, our predicate partition criteria follow the settings in Motifs (Zellers et al., 2018), as shown in Tab.8. This division strategy was introduced as early as 2018 and has been widely adopted in subsequent studies (Zhou et al., 2020; Yang et al., 2021; Tang et al., 2020). However, almost all existing works attempt to directly disentangle geometric and non-geometric predicates. Due to the predicate representation ambiguity arising from spatial co-occurrence, these methods often generate scene graphs that are factually correct yet semantically shallow. To address this problem, we propose to process the two types synergistically.

### C.3. Additional Quantitative Analysis

In the main text, we have demonstrated the effectiveness and versatility of our approach through detailed comparisons with various existing SOTA two-stage models. To further prove the superiority of our method, we re-conduct our experiments using Motif (Zellers et al., 2018) as the backbone, comparing our model directly with other methods that employ the same backbone under the exact same settings.

The experimental results are shown in Tab.9. It is worth noting that BGNN (Li et al., 2021), which we employed in the main paper, is also a widely recognized and open-source backbone in the field of SGG. The experiments in this section utilize Motif (Zellers et al., 2018) as the backbone solely to ensure a fair comparison. As shown in this table, our DS-Net demonstrates competitive performance across all metrics, validating the effectiveness of our method.

*Table 9.* The performance of state-of-the-art SGG models on three SGG tasks with graph constraints setting on mR@50/100, R@50/100 and M@50/100 on the VG dataset. The **best** methods are marked according to formats.

| Models | PredCls | | | SGCls | | | SGGen | | |
|---|---|---|---|---|---|---|---|---|---|
| | mR@50/100 | R@50/100 | M@50/100 | mR@50/100 | R@50/100 | M@50/100 | mR@50/100 | R@50/100 | M@50/100 |
| Motif (backbone) (Zellers et al., 2018) | 15.5/16.8 | **66.0/67.9** | 40.7/42.3 | 9.0/9.5 | 39.1/39.9 | 24.0/24.7 | 7.2/8.5 | **32.1/36.9** | 19.6/22.7 |
| TransRwt (Zhang et al., 2022) | 35.8/39.1 | 48.6/50.5 | 42.1/43.7 | **21.5**/22.8 | 29.4/30.2 | 24.1/24.8 | 15.8/18.0 | 23.5/27.2 | 16.9/19.6 |
| PPDL (Li et al., 2022b) | 32.2/33.3 | 47.2/47.6 | 39.7/40.4 | 17.5/18.2 | 28.4/29.3 | 22.9/23.7 | 11.4/13.5 | 21.2/23.9 | 16.3/18.7 |
| NICE (Li et al., 2022a) | 29.9/32.3 | 55.1/57.2 | 42.5/44.7 | 16.6/17.9 | 33.1/34.0 | 24.8/25.9 | 12.2/14.4 | 27.8/31.8 | 20.0/23.1 |
| EICR (Min et al., 2023) | 34.9/37.0 | 55.3/57.4 | 45.1/47.2 | 20.8/21.8 | 34.5/35.4 | 27.6/28.6 | 15.5/18.2 | 27.9/32.2 | 22.4/25.2 |
| PSCV (Zhou et al., 2022) | 33.2/35.2 | -/- | -/- | 20.8/21.9 | 32.5/33.6 | 26.6/27.7 | 10.1/13.9 | -/- | -/- |
| CFA (Li et al., 2023) | 35.7/38.2 | 54.1/56.6 | 44.9/47.4 | 17.0/18.4 | 34.9/36.1 | 25.9/27.2 | 13.2/15.5 | 24.7/31.8 | 18.9/23.6 |
| NICEST (Li et al., 2024d) | 29.5/31.6 | 59.1/61.0 | 44.3/46.3 | 15.7/16.5 | 34.4/35.2 | 25.0/25.8 | 10.4/12.4 | 28.0/32.4 | 19.2/22.4 |
| CAModule (Liu et al., 2025) | 36.7/39.3 | 59.8/63.4 | 48.2/51.3 | 21.1/**24.7** | 36.8/38.2 | 28.9/**31.4** | 16.3/18.2 | 29.1/32.7 | 22.7/25.4 |
| DS-Net | **38.4/40.2** | 62.5/63.6 | **50.4/51.9** | 19.9/21.4 | **39.9/40.4** | **29.9**/30.9 | **16.9/18.2** | 31.3/34.2 | **24.1/26.2** |

## C.4. Qualitative results

For the qualitative experiments of our DS-Net, we first visualize the spatial activation maps of the latent parts for the "girl" and "bat" in Fig.3. As observed, the model pays attention to specific regions of the girl (e.g., "head", "hands") and the bat (e.g., "handle", "face"). And as shown in Fig.4, our method accurately identifies not only geometric predicates but also non-geometric ones, demonstrating the effectiveness of our approach.

