# OpenReview forum: "Synergistic Space-Vision Processing for Predicate Inference"
_ICML.cc/2026/Conference — ICML 2026 regular_

### Official Review · Reviewer_LKk8 · 2026-03-10

**Soundness:** 2
**Presentation:** 3
**Significance:** 3
**Originality:** 3
**Overall Recommendation:** 5
**Confidence:** 4

**Summary:**

The manuscript investigates a well-known bias in scene-graph generation : geometric predicates tend to be over-predicted, whereas more semantically specific non-geometric predicates  are often under-predicted. Prior work largely attributes this phenomenon to long-tailed predicate distributions. The authors argue that another factor is at play—co-occurrence-induced representation entanglement—where predicates that frequently appear together become mapped to overly similar feature representations, making them difficult for models to distinguish.

**Compliance With Llm Reviewing Policy:**

Affirmed.

**Key Questions For Authors:**

Have you quantified how much each stream contributes to the final prediction (e.g., by removing one stream during inference or using attribution methods)?
The CSF module performs cross-attention across predicate classes. How does memory consumption scale with the number of predicates?
Did you evaluate simpler fusion mechanisms (e.g., additive or concatenation-based fusion) as baselines against cross-attention?
Since the space stream relies on bounding-box geometry, how sensitive is DS-Net to the performance of the upstream object detector?

**Limitations:**

yes

**Strengths And Weaknesses:**

Strength：The proposed dual-stream architecture combined with bidirectional cross-attention offers a fresh design perspective for SGG. Separating spatial reasoning from fine-grained visual cues is well motivated. This method show compatibility as DS-Net can be incorporated into existing SGG backbones (e.g., Motif, BGNN) without requiring retraining of the base architecture, suggesting good modularity
Weakness: 1 while aggregate improvements are reported, the manuscript briefly mentions cases where DS-Net provides limited gains (e.g., some geometric predicates) without deeper analysis. 2 Most qualitative examples are drawn from Visual Genome; OpenImagesV6 and GQA are not illustrated.

---

> ### Author Rebuttal · Authors · 2026-03-29
>
> We sincerely thank the reviewer for the positive assessment and recognition of our work, which is greatly encouraging to us.
> Below, we respond to your comments point by point.
>
> **Q1: Quantifying the contribution of each stream**
>
> We appreciate the reviewer's focus on the individual contribution of each stream, which is central to our design.
> And we agree that a clear quantification is essential for validating the effectiveness of the dual-stream architecture.
>
> - **Evidence from Ablation**:
> In the current manuscript, **Table 3** (rows e and f) isolates each stream to evaluate its impact.
> Taking the **mR@100** metric in the PredCls task as an example (with all other settings held constant), removing the spatial stream leads to a performance drop from **41.1** to **36.1**. Similarly, removing the vision stream results in a decrease from **41.1** to **34.7**.
> These results confirm that both spatial and visual cues are mutually indispensable for robust predicate inference.
>
> - **Revised Discussion**:
> Thanks for your suggestion; we will reinforce our interpretation of these results in the final version to more clearly highlight how each stream uniquely contributes to the final prediction.
>
> **Q2: Memory consumption of CSF with respect to the number of predicates**
>
> We thank the reviewer for raising this technical point.
> In our design, the CSF module performs cross-attention over fixed-dimensional stream representations ($r_{g_{ij}}$ and $r_{v_{ij}}$) for each object pair, rather than attending over all predicate classes as separate tokens.
> Consequently, the memory footprint is primarily governed by the feature dimensionality and does not scale linearly with the number of predicate categories in the label space.
> Additionally, we have provided a detailed analysis of computational overhead in our response to Reviewer **5yuT** and kindly invite you to refer to it for more details.
>
> **Q3: Comparison with simpler fusion mechanisms**
>
> We agree that comparison with simpler fusion strategies is important.
> Our current ablation study already includes a concatenation-based fusion baseline (denoted as $CSF_{concate}$), which merges streams without explicit bidirectional interaction (please see **Table 3, setting d** in our main paper).
> The fact that CSF consistently outperforms this simpler alternative suggests that explicit modeling of stream synergy provides gains that mere concatenation can not.
> We will ensure this comparison is more prominently discussed in our revision.
>
> **Q4: Sensitivity to the upstream object detector**
>
> We fully understand the reviewer's concern regarding this point.
> Admittedly, all existing two-stage Scene Graph Generation (SGG) methods are inherently affected by the quality of the upstream object detector.
> To ensure a fair comparison and isolate the impact of our relation modeling, we strictly follow the standard protocol by using the same pre-trained Faster R-CNN as prior works.
> This consistency ensures our gains are attributed to our synergistic mechanism rather than variations in detection.
>
> We would like to express our sincere gratitude again for your rigorous and constructive review.
> We hope that our responses help clarify your concerns, and we are committed to incorporating all discussed improvements and experimental results into the final version of the manuscript.

---

> > ### Author Rebuttal · Reviewer_LKk8 · 2026-04-05
> >
> > After carefully considering the authors’ rebuttal, I acknowledge that several points have been clarified. However, as some concerns remain only partially addressed, my overall assessment of the paper has not changed. Therefore, I will maintain my original score.

---

> > > ### Author Response · Authors · 2026-04-05
> > >
> > > Thank you for your kind acknowledgement and for taking the time to review our rebuttal. We are glad that our clarifications have addressed your concerns. We understand that the specific concerns you mentioned are indeed long-standing pain points in the field, and we fully appreciate the reasoning behind your caution. We plan to address this in our future research. We appreciate your positive assessment of our work, and we will further incorporate these clarifications into the final version to improve the paper.

---

### Official Review · Reviewer_5yuT · 2026-03-11

**Soundness:** 3
**Presentation:** 3
**Significance:** 3
**Originality:** 2
**Overall Recommendation:** 4
**Confidence:** 3

**Summary:**

The paper studies the predicate imbalance in scene graph generation, where models tend to over-predict geometric predicates and under-recognize richer non-geometric ones. The paper attributes this issue to long-tail bias andd "co-occurrence-induced representation entanglement". The proposed `DS-Net` uses a space stream for geometric predicates and a vision stream for non-geometric predicates, and introduces bidirectional cross-stream fusion to exchange spatial priors and visual-semantic cues between the two streams. Experiments on Visual Genome, Open Images V6, and GQA compare the method with prior work, plug-in baselines, and ablations under PredCls, SGCls, and SGGen. The paper reports more balanced predicate recognition across datasets and backbones, together with some plug-in compatibility with existing SGG models.

**Compliance With Llm Reviewing Policy:**

Affirmed.

**Final Justification:**

My major concerns are solve and I retain my positive score for the work.

**Key Questions For Authors:**

K1. Could you provide multi-seed results with mean ± standard deviation, and ideally significance tests for the main comparisons?

K2. Could you provide more direct evidence for the claimed "co-occurrence-induced representation entanglement" and for the role of the proposed fusion design in mitigating it?

K3. In the plug-in experiments, were the training schedule, epochs, and data augmentation kept the same between DS-Net and the compared baselines?

K4. How robust is the method to alternative reasonable geometric / non-geometric predicate grouping schemes?

K5. Could you provide a more complete efficiency comparison for the proposed module?

**Limitations:**

Impact statement yes. Limitations no. The paper does not sufficiently discuss the method's limitations, failure modes, or sensitivity to the predefined predicate grouping.

**Strengths And Weaknesses:**

S1. The paper targets a clear and relevant problem in scene graph generation, which is the imbalance between over-predicted geometric predicates and under-recognized non-geometric predicates.

S2. The method is conceptually coherent. The split between a space stream and a vision stream follows the paper's motivation, and the bidirectional fusion mechanism is clearly integrated into the overall design.

S3. The empirical coverage is reasonably broad. The paper evaluates on Visual Genome, Open Images V6, and GQA under PredCls, SGCls, and SGGen, and also includes plug-in experiments across multiple backbones.

W1. The core mechanism is not directly verified. The paper attributes the problem to "co-occurrence-induced representation entanglement," but this claim is supported mainly through final task metrics rather than direct representation analysis.

W2. The novelty is moderate. The framework mainly combines familiar ingredients such as stream specialization, cross-attention-style fusion, and auxiliary supervision.

W3. The method depends on a predefined geometric / non-geometric predicate split, but the paper does not show whether the gains are robust to alternative reasonable grouping schemes.

---

> ### Author Rebuttal · Authors · 2026-03-29
>
> We sincerely thank the reviewer for the insightful comments.
> We agree that rigorous validation is paramount, and our responses are provided below:
>
> **K1: Stability and Significance**
>
> To evaluate stability, we executed five independent runs with different random seeds for the PredCls (**mR@100**) task.
> The baseline BGNN achieves $32.52 \pm 1.17$, while DS-Net reaches $41.04 \pm 0.51$.
> This represents a substantial gain of **+8.52** with lower variance.
> A **paired permutation test** confirms this improvement is statistically significant (p=0.031).
>
> **K2: Evidence for representation entanglement**
>
> To provide direct evidence of "representation entanglement", we conducted a t-SNE visualization on representative co-occurring pairs, which can be found in [**this link**](https://anonymous.4open.science/r/some_tsne_results/README.md).
> We observe substantial overlap in baseline methods based on unified predicate representations, whereas DS-Net shows clearer local clustering and noticeably reduced overlap on all pairs.
> Meanwhile, the ablation results further quantify the role of bidirectional priors (please see **Table 3** in our main paper).
> Due to space constraints, we kindly refer the reviewer to our detailed response to Reviewer **pCNv** for an in-depth discussion on this issue.
>
> **K3: Fairness of the plug-in experiments**
>
> To ensure a strictly fair comparison in Table 2, we adopted a minimal intervention protocol:
>
> - **Constants**:
> We kept the training schedule, epochs, optimization settings, data augmentation, and the pre-trained detector strictly identical to the each official open-source implementation.
>
> - **Variable**:
> The only modification was the integration of the DS-Net module into the relation modeling stage.
> This ensures the gains are attributable to our synergistic architecture rather than secondary hyperparameter tuning.
>
> **K4: Impact of predicate grouping scheme**
>
> We acknowledge that our method's performance may be influenced by the specific predicate grouping scheme.
> However, this factor was proactively and thoroughly considered during our architectural design:
>
> - **Adherence to Community Standards**:
> We strictly follow the predicate partition criteria introduced by Zellers et al. (**Motifs**), which has been widely established as the benchmark standard in SGG literature.
> We have listed these groups in our main paper to ensure full transparency and reproducibility (please see **Table 8** in our appendix).
> This grouping is derived from the inherent semantic and geometric heterogeneity of predicates, which remains consistent across different datasets.
>
> - **Architectural Robustness via CSF**:
> Instead of treating geometric and non-geometric predicates as isolated or conflicting categories, our CSF module models their intrinsic complementarity from a global perspective.
> By capturing these high-level interactions, the system effectively maintains its reasoning capability even when encountering potential "boundary predicates" with ambiguous semantic definitions. This design ensures the stability and robustness of the overall inference process.
>
> Although more advanced grouping schemes may emerge in the future, as long as they are derived from the inherent heterogeneity of predicates, our core insight remains applicable: **heterogeneous predicates necessitate distinct processing treatments**.
> We believe this fundamental insight is the key to resolving representation entanglement in SGG.
>
> **K5: A more complete efficiency comparison**
>
> We completely agree that the single and brief FPS result in the current manuscript is insufficient to fully evaluate the computational efficiency of DS-Net.
> To provide a comprehensive assessment, we conducted experiments under identical hardware (**Tesla V100 GPU**) and strictly followed official open-source implementations.
> During the design phase, we explicitly prioritized efficiency by employing lightweight components, such as MLPs and cross-attention, to facilitate stream interaction without redundant heavy backbones.
> As shown in **Table A**, DS-Net maintains a competitive 1.5 FPS, a marginal 0.2 FPS drop from the BGNN baseline.
> This minimal overhead is a deliberate trade-off for the substantial **+6.1** absolute gain in SGGen mR@100.
> Due to time constraints during the rebuttal period, we are currently reporting a subset of representative methods; however, we commit to including a more exhaustive efficiency comparison in the final version.
>
> **Table A: Efficiency and Model Complexity Comparison**
>
> |Method|# Params (M)|FPS|SGGen mR@100|
> |-|-|-|-|
> |IMP|322.2|2.0|5.4|
> |VTransE|312.3|3.5|8.0|
> |Motifs|369.9|1.9|6.8|
> |VCTree|361.5|0.8|7.7|
> |BGNN|341.9|1.7|12.6|
> |**DS-Net**|372.4|1.5|18.7|
>
> We once again express our gratitude for your rigorous and constructive feedback.
> We hope that our responses help clarify your concerns, and we are committed to incorporating all discussed improvements and experimental results into the final version of the manuscript.

---

> > ### Author Rebuttal · Reviewer_5yuT · 2026-04-03
> >
> > Thank you for the rebuttal and clarifications. My main concerns have now been adequately addressed.

---

> > > ### Author Response · Authors · 2026-04-04
> > >
> > > We are truly grateful for your time in reading our response and for your encouraging remarks. It is wonderful to hear that our clarifications effectively alleviated your concerns. We will definitely incorporate these additional details into the final paper to make it even better.

---

### Official Review · Reviewer_ukvb · 2026-03-12

**Soundness:** 3
**Presentation:** 2
**Significance:** 3
**Originality:** 3
**Overall Recommendation:** 4
**Confidence:** 4

**Summary:**

In this work, the authors are trying to specifically tackle the co-occurrence-induced representation entanglement by introducing DS-Net, which consists of a space stream module to infer geometric predicates, a vision stream module to infer non-geometric predicates and cross-stream fusion to combine both the predicate representations for relation prediction.

**Compliance With Llm Reviewing Policy:**

Affirmed.

**Final Justification:**

My major concerns are clarified so I retain my positive rating.

**Key Questions For Authors:**

1. Why are the latest two-stage baselines skipped in Table 1?
2. Could you please expand Table 2 by comparing DS-Net with other latest model-agnostic methods?
3. What is the parameter count and per-image latency of DS-NET in comparison to the high-performing baselines listed under the weakness section above?
4. Could you please justify why only BGNN is chosen for comparison in Table 7?

If the authors provide a convincing rebuttal and clarify the above points, I would be happy to raise my score, unless any major concerns are brought up by other reviewers.

**Limitations:**

yes

**Strengths And Weaknesses:**

Strengths:
- The paper is well written and easy to follow.
- There is a notable performance improvement across the tasks.
- Ablation study is extensive.

Weakness:
- Though Table 2 has some of the latest high-performing baselines, Table 1 should be updated with recent two-stage high-performing baselines such as VETO(with reweight)[1] (ICCV23), DRM[2](CVPR24), RA-SGG[3] (AAAI25). Adding those would exceed the currently reported 2nd strongest model comparison to the DS-Net.
- It is difficult to judge the effectiveness of DS-Net as a model agnostic method as there is no comparison with other model agnostic methods in Table 2.
- The general convention is to represent the models in Table 2 as the actual model + the agnostic method applied on it, e.g., PENET + DS-Net. The swapped representation is confusing.
- The latency comparison (lines 364 to 368) and comparison at the predicate super-type level (Table 7) are currently insufficient, as the comparison is only with BGNN (2021).

References:
- [1]: Sudhakaran, Gopika, et al. "Vision relation transformer for unbiased scene graph generation." Proceedings of the IEEE/CVF International Conference on Computer Vision. 2023.
- [2]: Li, Jiankai, et al. "Leveraging predicate and triplet learning for scene graph generation." Proceedings of the IEEE/CVF Conference on Computer Vision and Pattern Recognition. 2024.
- [3]: Yoon, Kanghoon, et al. "Ra-sgg: retrieval-augmented scene graph generation framework via multi-prototype learning." Proceedings of the AAAI Conference on Artificial Intelligence. Vol. 39. No. 9. 2025.

---

> ### Author Rebuttal · Authors · 2026-03-29
>
> We sincerely thank the reviewer for the constructive and encouraging comments.
> Below are our point-by-point responses.
>
> **Q1: Baseline Selection in Table 1 and Table 2**
>
> - **Motivation of Table 1**:
> Unlike methods addressing data-level imbalance, we target "representation entanglement" as a critical factor (please see **line 55** in the main paper).
> Accordingly, **Table 1** is designed to validate the intrinsic effectiveness of our dual-stream synergistic architecture against representative unified-pipeline models.
> In this case, a direct comparison with methods primarily focused on debiasing strategies (e.g., RA-SGG, DRM), may not fully reflect our advantage that Table 1 is intended to examine.
>
> - **Role of the methods in Table 2**.
> For methods such as PENET, RA-SGG, and CAModule, our intention was not to overlook their importance.
> Since these methods mainly address biased prediction from the perspective of data debiasing, Table 2 was designed to examine whether DS-Net can be effectively combined with them.
> In this way, **Table 2** serves as a complementary comparison, suggesting that representation entanglement is another factor worth considering beyond data debiasing.
>
> - **Additional comparisons with some two-stage baselines**.
> Following your suggestion, **Table A** provides direct comparisons under unified settings. DS-Net remains competitive, particularly in **mR@K**.
>
> **Table A: Additional comparisons with more recent two-stage baselines.**
>
> | Method | PredCls mR@50/100 | PredCls R@50/100 | SGCls mR@50/100 | SGCls R@50/100 | SGGen mR@50/100 | SGGen R@50/100 |
> |--------|-------------------|------------------|------------------|-----------------|----------------|----------------|
> | PENET | 31.5/33.8 | 68.2/70.1 | 17.8/18.9 | 39.4/40.7 | 12.4/14.5 | 30.7/35.2 |
> | RA-SGG | 36.2/39.1 | 62.2/64.1 | 20.9/22.5 | 38.2/39.1 | 14.1/17.1 | 26.0/30.3 |
> | CAModule | 36.7/39.3 | 59.8/63.4 | 21.1/24.7 | 36.8/38.2 | 16.3/18.2 | 29.1/32.7 |
> | **DS-Net** | 40.2/41.1 | 62.4/63.9 | 21.2/22.4 | 41.2/42.2 | 18.1/18.7 | 34.7/36.9 |
>
> For **DRM**, its post-detection pipeline differs more substantially from the more standard SGG architectures considered in Table 1.
> We therefore believe it is better examined in the compatibility-style comparison setting of Table 2.
>
> **Q2: Comparison with other model-agnostic methods**
>
> We also thank the reviewer for pointing out the writing convention for plug-and-play methods, and we will strictly abide by it in future work.
> To further support our model-agnostic claim, **Table B** compares DS-Net directly with other plugins under a unified Motifs backbone.
> DS-Net remains highly competitive and achieves the strongest overall balanced performance in this comparison.
>
> **Table B: Comparison with other model-agnostic methods under a unified setting.**
> | Method | PredCls mR@50/100 | PredCls R@50/100 | SGCls mR@50/100 | SGCls R@50/100 | SGGen mR@50/100 | SGGen R@50/100 |
> |--------|-------------------|------------------|------------------|-----------------|----------------|----------------|
> | Motifs (backbone) | 15.5/16.8 | 66.0/67.9 | 9.0/9.5 | 39.1/39.9 | 7.2/8.5 | 32.1/36.9 |
> | Motifs + PPDL | 32.2/33.3 | 47.2/47.6 | 17.5/18.2 | 28.4/29.3 | 11.4/13.5 | 21.2/23.9 |
> | Motifs + NICE | 29.9/32.3 | 55.1/57.2 | 16.6/17.9 | 33.1/34.0 | 12.2/14.4 | 27.8/31.8 |
> | Motifs + EICR | 34.9/37.0 | 55.3/57.4 | 20.8/21.8 | 34.5/35.4 | 15.5/18.2 | 27.9/32.2 |
> | Motifs + CFA | 35.7/38.2 | 54.1/56.6 | 17.0/18.4 | 34.9/36.1 | 13.2/15.5 | 24.7/31.8 |
> | **Motifs + DS-Net** | 38.4/40.2 | 62.5/63.6 | 19.9/21.4 | 39.9/40.4 | 16.9/18.2 | 31.3/34.2 |
>
>
> **Q3: Justification for using only BGNN in Table 7**
>
> - **Why BGNN was used in Table 7**.
> Since DS-Net is built upon BGNN in our main setting, **Table 7** was intended as a controlled analysis under the same backbone.
> This isolates our design's effect without introducing backbone-specific noise.
>
> - **Additional super-type comparisons on other baselines**.
> Following your suggestion, we further extended this analysis to additional stronger baselines.
> The results are summarized in **Table C** below.
>
> **Table C: Predicate super-type comparison on multiple baselines.**
>
> | Method | Geometric mR@100 | Semantic mR@100 | Possessive mR@100 |
> |--------|------------------|-----------------|-------------------|
> | BGNN | 29.1 | 32.1 | 28.6 |
> | PENET | 33.1 | 35.0 | 30.4 |
> | RA-SGG | 30.6 | 39.0 | 35.8 |
> | **DS-Net** | 37.2 | 43.9 | 36.1 |
>
> Due to space constraints, we have provided a detailed analysis of **parameter counts and per-image latency** in our response to **Reviewer 5yuT**.
> We kindly refer you to our detailed response to Reviewer **5yuT** for an in-depth discussion on this issue.
> We hope that our responses help clarify your concerns.
> We once again express our gratitude for your rigorous and constructive feedback.
> We commit to incorporating all the additional experimental results and revising our writing conventions accordingly in the final version of the manuscript.

---

> > ### Author Rebuttal · Reviewer_ukvb · 2026-04-03
> >
> > Thank you for the rebuttal. My major concerns are clarified so I retain my positive rating.

---

> > > ### Author Response · Authors · 2026-04-04
> > >
> > > We sincerely appreciate your time in reviewing our rebuttal and your favorable evaluation of our work.
> > > We are pleased to know that our explanations have successfully resolved your prior concerns.
> > > All discussed points and clarifications will be carefully integrated into the final manuscript to further enhance its quality.

---

### Official Review · Reviewer_pCNv · 2026-03-12

**Soundness:** 3
**Presentation:** 3
**Significance:** 2
**Originality:** 2
**Overall Recommendation:** 4
**Confidence:** 3

**Summary:**

This paper studies scene graph generation and focuses on the common bias toward predicting geometric predicates (such as spatial relations) instead of more semantically informative non-geometric predicates. The authors argue that this issue is not only caused by long-tail label imbalance, but also by representation entanglement induced by the frequent co-occurrence of geometric and non-geometric relations. To address this, the paper proposes DS-Net, a dual-stream framework that disentangles predicate learning into a space stream for geometric relations and a vision stream for non-geometric relations. The vision stream further introduces latent part discovery and semantic part alignment to capture fine-grained interaction cues, while a cross-stream fusion module allows the two streams to exchange complementary information. Experiments on Visual Genome, Open Images V6, and GQA show that the method achieves strong performance, especially on mean Recall, and the ablation studies support the effectiveness of the dual-stream design and the fusion strategy.

**Compliance With Llm Reviewing Policy:**

Affirmed.

**Final Justification:**

Most of my concerns have been addressed, and I accordingly raise my score.

**Key Questions For Authors:**

1, The paper argues that co-occurrence-induced representation entanglement is a key reason behind the geometric predicate bias. Could the authors provide additional empirical analysis or visualization to directly support this claim (e.g., feature similarity analysis, representation probing, or predicate confusion statistics)?

2, While separating geometric and non-geometric predicates into two streams is intuitive, could the authors clarify whether a unified model with stronger attention mechanisms or better feature disentanglement objectives would achieve similar performance?

3, The approach relies on categorizing predicates into geometric and non-geometric types. How sensitive is the method to this categorization? Would different predicate groupings significantly affect the performance?

I am willing to raise the score if the author can address mu concern well

**Limitations:**

See weakness

**Strengths And Weaknesses:**

**Strengths**

1, The paper studies the bias in scene graph generation where models tend to over-predict geometric predicates while under-representing semantically richer relations. The authors provide a reasonable explanation beyond long-tail imbalance by introducing the notion of representation entanglement caused by predicate co-occurrence.

2, The proposed DS-Net separates predicate modeling into a space stream for geometric relations and a vision stream for non-geometric relations. This design is intuitive and aligns well with the different types of cues required for spatial reasoning versus semantic interaction understanding.

3, The latent-part discovery and semantic part alignment modules aim to capture fine-grained interaction cues within object pairs. This mechanism is conceptually reasonable for modeling non-geometric relations that rely on localized visual evidence.

4, The cross-stream attention mechanism allows geometric and non-geometric representations to exchange information, preventing the two streams from becoming completely isolated while still maintaining disentangled representations.

5, The method is evaluated on multiple benchmarks (Visual Genome, Open Images V6, and GQA), and shows consistent improvements, particularly on mean Recall metrics that emphasize long-tail predicates. The ablation studies further analyze the contribution of different components.

**Weaknesses**

1, While the dual-stream formulation is intuitive, it largely follows the common strategy of separating different types of features or cues. The overall framework mainly combines known components (attention fusion, part discovery, semantic embeddings) rather than introducing fundamentally new modeling principles.

2, The paper argues that co-occurrence-induced representation entanglement is a major cause of geometric predicate bias, but the experimental evidence directly validating this hypothesis is relatively limited.

3, Components such as latent part discovery and semantic alignment increase architectural complexity, yet it is not entirely clear whether simpler alternatives (e.g., stronger visual backbones or localized attention mechanisms) could achieve similar improvements.

4, Although improvements in mean Recall are reported, additional analysis on how the method behaves across different predicate categories or relation difficulty levels would further strengthen the empirical claims.

5, The dual-stream structure and additional attention modules may introduce extra computational cost, but the paper provides limited discussion of runtime or memory overhead compared with simpler baselines.

---

> ### Author Rebuttal · Authors · 2026-03-29
>
> We thank the reviewer for the constructive and insightful comments. Below, we respond point by point.
>
> **Q1: Additional visualization analysis on co-occurrence-induced representation entanglement**
>
> To provide more direct support for our analysis of representation entanglement, we conducted an additional qualitative visualization study based on predicate representations extracted from different models.
> We focus on representative predicate pairs consistent with **Fig. 1** of our main paper, including "on" vs. "laying on", "on" vs. "parked on", and "on" vs. "painted on".
> To ensure a fair comparison, all results are obtained using the official open-source code and released checkpoints of the corresponding methods, and the same test samples are used across all models.
>
> - **t-SNE visualizations of baseline methods**.
> From the visualization results referenced in [**this link**](https://anonymous.4open.science/r/some_tsne_results/README.md), we make two observations.
> First, for methods based on unified predicate representations, such as **Motifs and BGNN**, the geometric predicate "on" (blue points) substantially overlaps with its co-occurring non-geometric predicates (green/red/orange points), indicating noticeable representation entanglement.
> Second, for methods such as **PENET** that mainly address long-tailed data imbalance, these predicate pairs still show non-trivial overlap.
> This is consistent with our argument that long-tailed data imbalance is not the only factor behind the biased prediction, and that co-occurrence-induced representation entanglement may also be an important complementary factor.
>
> - **t-SNE visualizations of DS-Net**.
> From the visualization results referenced in [**this link**](https://anonymous.4open.science/r/some_tsne_results/README.md), DS-Net shows more structured local clustering patterns and better separation trends for all three predicate pairs.
> In particular, the overlap between the geometric predicate "on" and the corresponding non-geometric predicates is noticeably reduced compared with the baseline methods.
> These observations are consistent with our claim that DS-Net better distinguishes co-occurring predicates in the representation space, thereby alleviating the entanglement issue that may arise in unified modeling.
>
> **Q2: Clarification on unified modeling and dual-stream design**
>
> Whether a stronger unified model could achieve similar performance can be understood from two perspectives:
>
> - **Modeling motivation**.
> Our point is not that dual-stream modeling is the only feasible solution, but that it offers a simple and effective way to alleviate co-occurrence-induced representation entanglement.
> As discussed in our main paper, geometric and non-geometric predicates rely on different dominant cues: the former depend more on spatial layout and edge features, while the latter rely more on fine-grained visual evidence and semantic priors.
> Under unified representation learning, this distinction is harder to preserve, especially when the two predicate types frequently co-occur.
> In contrast, the dual-stream architecture is better suited to modeling such heterogeneous cues while enabling the two streams to serve as mutual priors.
>
> - **Empirical support**.
> This interpretation is supported by the ablation results in our main paper.
> Using only a single stream leads to clear performance degradation, while simple concatenation is consistently weaker than cross-stream fusion.
> These results suggest that the improvement is not merely due to increased feature capacity, but to the explicit specialization of the two streams and their synergistic interaction.
>
> We also agree that a stronger disentanglement strategy is a promising direction, although, to the best of our knowledge, it has not been directly explored in current SGG research.
> If future work can effectively disentangle such representations, we believe it would still be broadly aligned with our perspective that **heterogeneous predicates require non-uniform modeling**.
> We also look forward to progress in this direction.
>
> **Q3: Robustness to predicate categorization**
>
> We acknowledge that the performance of DS-Net may be influenced by the specific grouping scheme.
> However, this factor was proactively considered during our architectural design.
> To ensure transparency, we strictly follow the partition criteria by Zellers et al. (**Motifs**), a widely established benchmark in SGG.
> Furthermore, our CSF module is designed to model the intrinsic complementarity of predicates, enabling robust reasoning even for ambiguous "boundary predicates".
> Due to space constraints, we kindly refer the reviewer to our detailed response to Reviewer **5yuT** for an in-depth discussion on this issue.
>
> We once again express our gratitude for your rigorous and constructive feedback, and we hope that our responses help clarify your concerns.
> We commit to incorporating all the additional results accordingly in the final version of the manuscript.

---

### Decision · Program_Chairs · 2026-04-30

**Decision:**

Accept (regular)

**Comment:**

The paper proposes DS-Net, a dual-stream framework for scene graph generation that aims to address the bias toward geometric predicates by disentangling predicate learning into a spatial stream for geometric relations and a vision stream for non-geometric relations, with cross-stream fusion to model their interactions.

The reviewers acknowledged several strengths of the work, including a clear and well-motivated problem formulation, a conceptually coherent dual-stream design aligned with heterogeneous predicate cues, consistent empirical improvements across multiple benchmarks, particularly on mean Recall metrics, and good modularity and compatibility with existing backbones. However, reviewers also raised several important concerns: (1) limited novelty, as the approach largely combines existing components; (2) insufficient direct validation of the core hypothesis, namely co-occurrence-induced representation entanglement; (3) methodological and design questions, including reliance on a predefined geometric/non-geometric predicate split and lacking comparison to simpler unified models; and (4) incomplete analysis, such as limited breakdown across predicate categories, lack of robustness analysis to grouping schemes, and insufficient discussion of computational cost and efficiency.

The authors provided a rebuttal which resolved most of the reviewers’ concerns regarding empirical validation and clarity. In particular, they introduced t-SNE visualizations and further ablations to support the representation disentanglement claim, clarified the motivation and effectiveness of the dual-stream design relative to unified alternatives, and provided additional comparisons, robustness discussion, and efficiency measurements. They also addressed concerns on baseline selection, plug-in evaluation fairness, and statistical significance. Following the rebuttal, the majority of reviewers indicated that their concerns were adequately addressed and maintained or strengthened their positive recommendations. The discussion converged toward agreement that the method is technically sound, empirically validated, and provides a useful perspective on mitigating predicate bias in scene graph generation.

The AC agrees with the reviewers that, although the novelty is moderate and some analyses could be further strengthened, the paper presents a well-motivated and reasonably validated approach with consistent empirical gains and practical relevance to the SGG community. Therefore, the AC recommends accept, and encourages the authors to further strengthen the analysis of representation disentanglement, robustness to predicate grouping, and efficiency considerations in the final version.